# CONSTRAINED MARKOV DECISION PROCESSES VIA BACKWARD VALUE FUNCTIONS

## ABSTRACT

Although Reinforcement Learning (RL) algorithms have found tremendous success in simulated domains, they often cannot directly be applied to physical systems, especially in cases where there are hard constraints to satisfy (e.g. on safety or resources). In standard RL, the agent is incentivized to explore any behavior as long as it maximizes rewards, but in the real world undesired behavior can damage either the system or the agent in a way that breaks the learning process itself. In this work, we model the problem of learning with constraints as a Constrained Markov Decision Process, and provide a new on-policy formulation for solving it. A key contribution of our approach is to translate cumulative cost constraints into state-based constraints. Through this, we define a safe policy improvement method which maximizes returns while ensuring that the constraints are satisfied at every step. We provide theoretical guarantees under which the agent converges while ensuring safety over the course of training. We also highlight computational advantages of this approach. The effectiveness of our approach is demonstrated on safe navigation tasks and in safety-constrained versions of MuJoCo environments, with deep neural networks.

## 1    INTRODUCTION

Reinforcement Learning (RL) provides a sound decision-theoretic framework to optimize the behavior of learning agents in an interactive setting (Sutton & Barto, 2018). Recently, the field of RL has found success in many high-dimensional domains, like video games, Go, robot locomotion and navigation. However, most of the success of RL algorithms has been limited to simulators, where the learning algorithm has the ability to reset the simulator. In the physical world, an agent will need to avoid harmful behavior (e.g. damaging the environment or the agent's hardware) while learning to explore behaviors that maximize the reward.

A few popular approaches for avoiding undesired behaviors for high-dimensional systems include reward-shaping (Moldovan & Abbeel, 2012), reachability-preserving algorithms (Mitchell, 2003; Eysenbach et al., 2017), state-level surrogate constraint satisfaction algorithms (Dalal et al., 2018), risk-sensitive algorithms (Tamar et al., 2013; Chow et al., 2015) and apprenticeship learning (Abbeel & Ng, 2004). There also exists model-based Bayesian approaches that are focused on imposing the constraints via the dynamics (such as classifying parts of state space as unsafe) and then using model predictive control to incorporate the constraints in the policy optimization and planning (Turchetta et al., 2016; Berkenkamp et al., 2017; Wachi et al., 2018; Koller et al., 2018). A natural way to model safety is via constraint satisfaction. A standard formulation for adding constraints to RL problems is the Constrained Markov Decision Process (CMDP) framework (Altman, 1999), wherein the environment is extended to also provide feedback on constraint costs. The agent must then attempt to maximize its expected return while also satisfying cumulative constraints.

A few algorithms have been proposed to solve CMDPs for high-dimensional domains with continuous action spaces - however they come with their own caveats. Reward Constrained Policy Optimization (Tessler et al., 2018) and Primal Dual Policy Optimization (Chow et al., 2015) do not guarantee constraint satisfaction during the learning procedure, only on the final policy. Constrained Policy Optimization (Achiam et al., 2017) provides monotonic policy improvement but is computationally expensive due to requiring a backtracking line-search procedure and conjugate gradient algorithm for approximating the Fisher Information Matrix. Lyapunov-based Safe Policy Optimization (Chow

et al., 2019) requires solving a Linear Program (LP) at every step of policy evaluation, although they show that there exists heuristics which can be substituted for the LP at the expense of theoretical guarantees.

In this work, we propose an alternate formulation for solving CMDPs that transforms trajectory-level constraints into localized state-dependent constraints, through which a safe policy improvement step can be defined. In our approach, we define a notion of Backward Value Functions, which act as an estimator of the expected cost collected by the agent so far and can be learned via standard RL bootstrap techniques. We provide conditions under which this new formulation is able to solve CMDPs without violating the constraints during the learning process. Our formulation allows us to define state-level constraints without explicitly solving a LP or the Dual problem at every iteration. Our method is implemented as a reduction to any model-free on-policy bootstrap based RL algorithm, both for deterministic and stochastic policies, and discrete and continuous action spaces. We provide the empirical evidence of our approach with Deep RL methods on various safety benchmarks, including 2D navigation grid worlds (Leike et al., 2017; Chow et al., 2018), and MuJoCo tasks (Achiam et al., 2017; Chow et al., 2019).

## 2 CONSTRAINED MARKOV DECISION PROCESSES

We write $\mathscr{P}(Y)$ for the set of probability distributions on a space $Y$. A Markov Decision Process (MDP) (Puterman, 2014) is a tuple $(\mathcal{X}, \mathcal{A}, \mathcal{P}, r, x_0)$, where $\mathcal{X}$ is a set of states, $\mathcal{A}$ is a set of actions, $r : \mathcal{X} \times \mathcal{A} \to [0, R_{MAX}]$ is a reward function, $\mathcal{P} : \mathcal{X} \times \mathcal{A} \to \mathscr{P}(\mathcal{X})$ is a transition probability function, and $x_0$ is a fixed starting state. For simplicity we assume a deterministic reward function and starting state, but our results generalize.

A *Constrained Markov Decision Process (CMDP)* (Altman, 1999) is a MDP with additional constraints that restrict the set of permissible policies for the MDP. Formally, a CMDP is a tuple $(\mathcal{X}, \mathcal{A}, \mathcal{P}, r, x_0, d, d_0)$, where $d : \mathcal{X} \to [0, D_{MAX}]$ is the cost function[1] and $d_0 \in \mathbb{R}^{\geq 0}$ is the maximum allowed cumulative cost. The set of *feasible* policies that satisfy the CMDP is the subset of stationary policies $\Pi_{\mathcal{D}} := \{\pi : \mathcal{X} \to \mathscr{P}(\mathcal{A}) \mid \mathbb{E}[\sum_{t=0}^{T} d(x_t) \mid x_0, \pi] \leq d_0\}$. We consider a finite time horizon $T$ after which the episode terminates. The expected sum of rewards following a policy $\pi$ from an initial state $x$ is given by the value function $V^{\pi}(x) = \mathbb{E}[\sum_{t=0}^{T} r(x_t, a_t) \mid \pi, x]$. Analogously, the expected sum of *costs* is given by the cost value function $V_{\mathcal{D}}^{\pi}(x) = \mathbb{E}[\sum_{t=0}^{T} d(x_t) \mid \pi, x]$. The RL problem in the CMDP is to find the feasible policy which maximizes expected returns from the initial state $x_0$, i.e.

$$\pi^* = \arg\max_{\pi \in \Pi_{\mathcal{D}}} V^{\pi}(x_0)$$

An important point to note about CMDPs is that, in the original formulation, the cost function depends on immediate states but the constraint is cumulative and thus depends on the entire trajectory.

In the case of MDPs, where a model of the environment is not known or is not easily obtained, it is still possible for the agent to find the optimal policy using Temporal Difference (TD) methods (Sutton, 1988). Broadly, these methods update the estimates of the value functions via bootstraps of previous estimates on sampled transitions (we refer the reader to Sutton & Barto (2018) for more information). In the on-policy setting, we alternate between estimating the state-action value function $Q^{\pi}$ for a given $\pi$ and updating the policy to be greedy with respect to the value function.

## 3 SAFE POLICY ITERATION VIA BACKWARD VALUE FUNCTIONS

Our approach proposes to convert the trajectory-level constraints of the CMDP into single-step state-wise constraints in such a way that satisfying the state-wise formulation will entail satisfying the original trajectory-level problem. The advantages of this approach are twofold: i) working with single-step state-wise constraints allows us to obtain analytical solutions to the optimization problem, and ii) the state-wise constraints can be defined via value-function-like quantities and can thus be estimated with well-studied value-based methods. The state-wise constraints are defined via *Backward Value Functions*, in Section 3.2, and in Section 3.3 we provide a safe policy iteration procedure which satisfies said constraints (and thus the original problem).

---

[1]Here the cost only depends on states and not state-action pairs.

## 3.1 BACKWARD MARKOV CHAIN

Unlike in traditional RL, in the CMDP setting the agent needs to take into account the constraints which it has accumulated so far in order to plan accordingly for the future. Intuitively, the accumulated cost so far can be estimated via the cost value function $V_{\mathcal{D}}$ running "backward in time". Before giving the details of our approach and formally introducing the Backward Value Functions, we review the main ideas, which are built upon the work of Morimura et al. (2010), who also considered time-reversed Markov chains but from the standpoint of estimating the gradient of the log stationary distribution; we extend these ideas to TD methods.

**Assumption 3.1** (Stationarity). The MDP is ergodic for any policy $\pi$, i.e., the Markov chain characterized by the transition probability $\mathcal{P}^\pi(x_{t+1}|x_t) = \sum_{a_t \in \mathcal{A}} \mathcal{P}(x_{t+1}|x_t, a_t)\pi(a_t|x_t)$ is irreducible and aperiodic.

Let $\mathcal{M}(\pi)$ denote the Markov chain characterized by transition probability $\mathcal{P}^\pi(x_{t+1}|x_t)$. The above assumption implies that there exists a unique stationary distribution $\eta^\pi$ associated with $\pi$, such that it satisfies: $\eta^\pi(x_{t+1}) = \sum_{x_t \in \mathcal{X}} \mathcal{P}^\pi(x_{t+1}|x_t)\eta^\pi(x_t)$. We abuse the notation and denote $\mathcal{P}^\pi(x_{t+1}, a_t|x_t) = \mathcal{P}(x_{t+1}|x_t, a_t)\pi(a_t|x_t)$.

According to Bayes' rule, the probability $q(x_{t-1}, a_{t-1}|x_t)$ of a previous state-action pair $(x_{t-1}, a_{t-1})$ leading to the current state $x_t$ is given by:

$$q(x_{t-1}, a_{t-1}|x_t) = \frac{\mathcal{P}(x_t|x_{t-1}, a_{t-1})Pr(x_{t-1}, a_{t-1})}{\sum_{x_{t-1} \in \mathcal{X}} \sum_{a_{t-1} \in \mathcal{A}} \mathcal{P}(x_t|x_{t-1}, a_{t-1})Pr(x_{t-1}, a_{t-1})}.$$

From Assumption 3.1, we have that $Pr(x_{t-1}, a_{t-1}) = \eta^\pi(x_{t-1})\pi(a_{t-1}|x_{t-1})$, and $\sum_{x_{t-1} \in \mathcal{X}} \sum_{a_{t-1} \in \mathcal{A}} \mathcal{P}(x_t|x_{t-1}, a_{t-1})Pr(x_{t-1}, a_{t-1}) = \eta^\pi(x_t)$. We denote the posterior $q(x_{t-1}, a_{t-1}|x_t)$ as **backward (or time-reversed)** probability $\overleftarrow{\mathcal{P}}^\pi(x_{t-1}, a_{t-1}|x_t)$, and we have:

$$\overleftarrow{\mathcal{P}}^\pi(x_{t-1}, a_{t-1}|x_t) = \frac{\mathcal{P}(x_t|x_{t-1}, a_{t-1})\eta^\pi(x_{t-1})\pi(a_{t-1}|x_{t-1})}{\eta^\pi(x_t)}$$
$$= \frac{\mathcal{P}^\pi(x_t, a_{t-1}|x_{t-1})\eta^\pi(x_{t-1})}{\eta^\pi(x_t)}. \tag{1}$$

The forward Markov chain, characterized by the transition matrix $\mathcal{P}^\pi(x_{t+1}|x_t)$, runs forward in time, i.e., it gives the probability of the next state in which the agent *will end up*. Analogously, a backward Markov chain is denoted by the transition matrix $\overleftarrow{\mathcal{P}}^\pi(x_{t-1}|x_t) = \sum_{a_{t-1} \in \mathcal{A}} \overleftarrow{\mathcal{P}}^\pi(x_{t-1}, a_{t-1}|x_t)$, and describes the state and action the agent *took* to reach the current state.

**Definition 3.1** (Backward Markov Chain). A backward Markov chain associated with $\mathcal{M}(\pi)$ is denoted by $\overleftarrow{\mathcal{B}}(\pi)$ and is characterized by the transition probability $\overleftarrow{\mathcal{P}}^\pi(x_{t-1}|x_t)$.

## 3.2 BACKWARD VALUE FUNCTION

We define the *Backward Value Function* (BVF) to be a value function running on the backward Markov chain $\overleftarrow{\mathcal{B}}(\pi)$. A BVF is the expected sum of returns or costs collected by the agent *so far*. We are mainly interested in maintaining estimates of the cumulative cost incurred at a state in order to express the total constraint state-wise.

We note that, since every Markov chain $\mathcal{M}(\pi)$ is ergodic by Assumption 3.1, the corresponding backward Markov chain $\mathcal{B}(\pi)$ is also ergodic (Morimura et al., 2010, Prop. B.1). In particular, every policy $\pi$ can reach the initial state via some path in the transition graph of the backward Markov chain. Thus, the backwards Markov chain are also finite-horizon for some $T_{\mathcal{B}}$, with $x_0$ corresponding to the terminal state. We define a finite-horizon Backward Value Function for cost as:

$$\overleftarrow{V}_{\mathcal{D}}^\pi(x_t) = \mathbb{E}_{\overleftarrow{\mathcal{B}}(\pi)} \left[ \sum_{k=0}^{T_{\mathcal{B}}} d(x_{t-k})|x_t \right]. \tag{2}$$

**Proposition 3.1** (Sampling). Samples from the forward Markov chain $\mathcal{M}(\pi)$ can be used directly to estimate the statistics of the backward Markov chain $\overleftarrow{\mathcal{B}}(\pi)$ (or the Backward Value Function). We

have:

$$\mathbb{E}_{\overleftarrow{\mathcal{B}}(\pi)}\left[\sum_{k=0}^{K} d(x_{t-k})|x_t\right] = \mathbb{E}_{\mathcal{M}(\pi)}\left[\sum_{k=0}^{K} d(x_{t-k})|x_t, \eta^\pi(x_{t-K})\right], \tag{3}$$

$$= \mathbb{E}_{\mathcal{M}(\pi)}\left[\sum_{k=0}^{K} d(x_{t+k})|x_{t+K}, \eta^\pi(x_t)\right],$$

where $\mathbb{E}_{\mathcal{M}(\pi)}$ and $\mathbb{E}_{\overleftarrow{\mathcal{B}}(\pi)}$ are expectations over the forward and backward chains respectively. The Equation (3) holds true even in the limit $K \to \infty$.

The proof is given in Appendix B.1. Using the above proposition, we get an interchangeability property that removes the need to sample from the backward chain. We can use the traditional RL setting and draw samples from the forward chain and still estimate the BVFs. Equation (2) can be written recursively as:

$$\overleftarrow{V}_{\mathcal{D}}^\pi(x_t) = \mathbb{E}_{\overleftarrow{\mathcal{B}}(\pi)}\left[d(x_t) + \overleftarrow{V}_{\mathcal{D}}^\pi(x_{t-1})\right].$$

In operator form, the above equation can also be written as:

$$(\overleftarrow{\mathcal{T}}^\pi \overleftarrow{V}_{\mathcal{D}}^\pi)(x_t) = \mathbb{E}_{x_{t-1} \sim \overleftarrow{P}^\pi}\left[d(x_t) + \overleftarrow{V}_{\mathcal{D}}^\pi(x_{t-1})\right]. \tag{4}$$

**Proposition 3.2** (Fixed point). For a policy $\pi$, the associated Backward Value Function vector, $\overleftarrow{V}^\pi$, satisfies $\lim_{k\to\infty}\left(\overleftarrow{\mathcal{T}}^\pi\right)^k \overleftarrow{V} = \overleftarrow{V}^\pi$ for every vector $\overleftarrow{V}$, and $\overleftarrow{V}^\pi$ is the unique solution of the equation $\overleftarrow{V}^\pi = \overleftarrow{\mathcal{T}}^\pi \overleftarrow{V}^\pi$.

The proof is given in Appendix B.2. The above proposition allows us to soundly extend the RL methods based on Bellman operators for the estimation of BVFs.

### 3.3 SAFE POLICY IMPROVEMENT VIA BVF-BASED CONSTRAINTS

With the Backward Value Function framework, the trajectory-level optimization problem associated with a CMDP can be rewritten in state-wise form. Recall that a feasible policy must satisfy the constraint:

$$\mathbb{E}_{\mathcal{M}(\pi)}\left[\sum_{k=0}^{T} d(x_k) \mid x_0\right] \le d_0.$$

Alternatively, for each timestep $t \in [0, T]$ of a trajectory:

$$\mathbb{E}\left[\sum_{k=0}^{t} d(x_k) \mid x_0, \pi\right] + \mathbb{E}\left[\sum_{k=t}^{T} d(x_k) \mid x_0, \pi\right] - \mathbb{E}\left[d(x_t) \mid x_0\right] \le d_0.$$

Via the identities $\mathbb{E}[\sum_{k=t}^{T} d(x_k) \mid x_0, \pi] \le \mathbb{E}_{x_t \sim \delta_{x_0}(P^\pi)^t}[V_{\mathcal{D}}^\pi(x_t)]$ and $\mathbb{E}[\sum_{k=0}^{t} d(x_k) \mid x_0, \pi] \le \mathbb{E}_{x_k \sim \delta_{x_0}(P^\pi)^t}[\overleftarrow{V}_{\mathcal{D}}^\pi(x_t)]$(derived in Appendix C)[2], we remark that the quantity on the LHS is less than the expectation over $k$-step trajectories of $\overleftarrow{V}_{\mathcal{D}}^\pi(x_t) + V_{\mathcal{D}}^\pi(x_t) - d(x_t)$. In other words, for each $t \in [0, T]$ :

$$\mathbb{E}_{\mathcal{M}(\pi)}\left[\sum_{k=0}^{T} d(x_k) \mid x_0\right] \le \mathbb{E}_{x_t \sim \delta_{x_0}(P^\pi)^t}\left[\overleftarrow{V}_{\mathcal{D}}^\pi(x_t) + V_{\mathcal{D}}^\pi(x_t) - d(x_t)\right] \le d_0.$$

These are the state-wise constraints that should hold at each step in a given trajectory - we refer to them as the **value-based constraints**. Satisfying the value-based constraints will automatically satisfy the given CMDP constraints.

---

[2] $\delta_{x_0}$ is a Dirac distribution at $x_0$, and $\delta_{x_0}(P^\pi)^t$ is the distribution of states at time $t$.

This formulation allows us to introduce a policy improvement step, which maintains a safe feasible policy at every iteration by using the previous estimates of the forward and backward value functions[3]. The policy improvement step is defined by a linear program, which performs a greedy update with respect to the current state-action value function subject to the value-based constraints:

$$\pi_{k+1}(\cdot|x) = \arg\max_{\pi \in \Pi} \left\langle \pi(\cdot|x), Q^{\pi_k}(x, \cdot) \right\rangle, \tag{SPI}$$

$$s.t. \ \left\langle \pi(\cdot|x), Q_{\mathcal{D}}^{\pi_k}(x, \cdot) \right\rangle + \overleftarrow{V}_{\mathcal{D}}^{\pi_k}(x) - d(x) \leq d_0, \quad \forall x \in \mathcal{X}.$$

Our first result is that the policies obtained by the policy improvement step will satisfy the safety constraints. We write $\mathrm{TV}(\cdot, \cdot)$ for the total variation metric between distributions.

**Theorem 3.1** (Consistent Feasibility). Assume that successive policies are updated sufficiently slowly, i.e. $\mathrm{TV}(\pi_{k+1}(\cdot|x), \pi_k(\cdot|x)) \leq \frac{d_0 - V_{\mathcal{D}}^{\pi_k}(x_0)}{2D_{\mathrm{MAX}}T^2}$.[4] Then the policy iteration step given by (SPI) is consistently feasible, i.e. if $\pi_k$ is feasible at $x_0$ then so is $\pi_{k+1}$.

It is also possible to consider larger neighbourhoods for updates of successive policies, but at the cost of everywhere-feasibility. For want of space, we present that result in Appendix D.

Next we show that the policy iteration step given by (SPI) leads to monotonic improvement.

**Theorem 3.2** (Policy Improvement). Let $\pi_n$ and $\pi_{n+1}$ be successive policies generated by the policy iteration step of (SPI). Then $V^{\pi_{n+1}}(x) \geq V^{\pi_n}(x) \ \forall x \in \mathcal{X}$. In particular, the sequence of value functions $\{V^{\pi_n}\}_{n \geq 0}$ given by (SPI) monotonically converges.

Proofs for Theorems 3.1 and 3.2 are given in Appendix D. Finding the sub-optimality gap (if any) remains an interesting question left for future work.

## 4 PRACTICAL IMPLEMENTATION CONSIDERATIONS

### 4.1 DISCRETE ACTION SPACE

In discrete action spaces, the problem in (SPI) can be solved exactly as a Linear Programming problem. It is possible to approximate its analytical solution by casting it into the corresponding entropy-regularized counterpart (Neu et al., 2017; Chow et al., 2018). The details of the closed form solution can be found in Appendix E.

Furthermore, if we restrict the set of policies to be deterministic, then it is possible to have an in-graph solution as well. The procedure then closely resembles the Action Elimination Procedure (Puterman, 2014, Chapter 6), where non-optimal actions are identified as being those which violate the constraints.

### 4.2 EXTENSION TO CONTINUOUS CONTROL

For MDPs with only state-dependent costs, Dalal et al. (2018) proposed the use of safety layers, a constraint projection approach, that enables action correction at each step. At any given state, an unconstrained action is selected and is passed to the safety layer, which projects the action to the nearest action (in Euclidean norm) satisfying the necessary constraints. We extend this approach to stochastic policies to handle the corrections for the actions generated by stochastic policies. When the policy is parameterized with a Gaussian distribution, then the safety-layer can still be used by projecting both the mean and standard-deviation vector to the constraint-satisfying hyper-plane[5]. In most cases, the standard-deviation vector is kept fixed or independent of the state (Kostrikov, 2018; Dhariwal et al., 2017), which allows us to formulate the problem as solving the following $L2$-projection of the mean of the Gaussian in Euclidean space. For $\mu_\pi(.; \theta)$, at any given state $x \in \mathcal{X}$,

---

[3]In general, it is not possible to obtain the expectation $\mathbb{E}_{x_t \sim \delta_{x_0}(P^\pi)^t}[\cdot]$ directly as it may be intractable to compute or we may not have access to the true transition distributions of the model. Thus, we sample a batch of transitions from the current policy and use them for the updates.

[4]This can be enforced, for example, by constraining iterates to a neighborhood $D(\pi, \pi_k) \leq \delta$.

[5]More information about this claim can be found in Appendix F.

the safety layer solves the following projection problem:

$$\underset{\mu}{\arg\min} \left[ \frac{1}{2} \left\| (\mu - \mu_\pi(x)) \right\|^2 \right],$$
$$\texttt{s.t.} \quad Q_\mathcal{D}^\pi(x, \mu) + \overleftarrow{V}_\mathcal{D}^\pi(x) - d(x) \leq d_0.$$

As shown in Dalal et al. (2018); Chow et al. (2019), if the constraints have linear nature then an analytical solution exists. In order to get a linearized version of the constraints (and simplify the projection), we can approximate the constraint with its first-order Taylor series at $\mu = \mu_\pi(x)$:

$$\underset{\mu}{\arg\min} \left[ \frac{1}{2} \left\| (\mu - \mu_\pi(x)) \right\|^2 \right], \tag{5}$$
$$\texttt{s.t.} \quad \overleftarrow{V}_\mathcal{D}^\pi(x) - d(x) + \underbrace{Q_\mathcal{D}^\pi(x, \mu_\pi(x)) + (\mu - \mu_\pi(x))^T (\nabla Q_\mathcal{D}^\pi(x, \mu)|_{\mu = \mu_\pi(x)})}_{\text{First order Taylor expansion}} \leq d_0.$$

The above objective function is positive-definite and quadratic, and the constraints are linear. Though this problem can be solved by an in-graph QP solver, there exists an analytical solution (see Appendix G):

**Proposition 4.1.** At a given state $x \in \mathcal{X}$, the solution to the Eq. (5), $\mu^*$ is:

$$\mu^* = \mu_\pi(x) - \lambda^*(x) \cdot g_{\mu, D}(x),$$
$$\text{where,} \qquad g_{\mu, \mathcal{D}}(x) = \nabla Q_\mathcal{D}^\pi(x, \mu)|_{\mu = \mu_\pi(x)},$$
$$\lambda^*(x) = \left( \frac{-(d_0 + d(x) - \overleftarrow{V}_\mathcal{D}^\pi(x) - Q_\mathcal{D}^\pi(x, \mu_\pi(x)))}{g_{\mu, \mathcal{D}}(x)^T g_{\mu, \mathcal{D}}(x)} \right)^+.$$

## 5 RELATED WORK

**Lagrangian-based methods:** Initially introduced in Altman (1999), more scalable versions of the Lagrangian based methods have been proposed over the years (Moldovan & Abbeel, 2012; Tessler et al., 2018; Chow et al., 2015). The general form of the Lagrangian methods is to convert the problem to an unconstrained problem via Langrange multipliers. If the policy parameters are denoted by $\theta$, then Lagrangian formulation becomes: $\min_{\lambda \geq 0} \max_\theta (L(\theta, \lambda) = \min_{\lambda \geq 0} \max_\theta [V^{\pi_\theta}(x_0) - \lambda(V_\mathcal{D}^{\pi_\theta}(x_0) - d_0))]$, where $L$ is the Lagrangian and $\lambda$ is the Lagrange multiplier (penalty coefficient). The main problems of the Lagrangian methods are that the Lagrangian multiplier is either a hyper-parameter (without much intuition), or is solved on a lower time-scale. That makes the unconstrained RL problem a three time-scale [6] problem, which makes it very difficult to optimize in practice. Another problem is that during the optimization, this procedure can violate the constraints. Ideally, we want a method that can respect the constraint throughout the training and not just at the final optimal policy.

**Lyapunov-based methods:** In control theory, the stability of the system under a fixed policy is computed using Lyapunov functions (Khalil, 1996). A Lyapunov function is a type of scalar potential function that keeps track of the energy that a system continually dissipates. Recently, Chow et al. (2018; 2019) provide a method of constructing the Lyapunov functions to guarantee global safety of a behavior policy using a set of local linear constraints. Their method requires the knowledge of $TV(\pi, \pi^*)$ to guarantee the theoretical claims. They substitute the ideally required Lyapunov function with an approximate solution that requires solving a LP problem at every iteration. For the practical scalable versions, they use a heuristic, a constant Lyapunov function for all states that only depends on the initial state and the horizon. While our methods also constructs state-wise constraints, there are two notable differences: a) our assumption only rely on the current policy candidate and the baseline policy, instead of the baseline and the optimal policy, b) our method does not require solving an LP at every update step to construct the constraint and as such the only approximation error that is introduced comes from the function approximation.

---

[6] Classic Actor Critic is two time-scale (Konda & Tsitsiklis, 2000), and adding a learning schedule over the Lagrangian makes it three time scale.

**Conservative Policy Improvement:** Constrained Policy Optimization (CPO) (Achiam et al., 2017) extends the trust-region policy optimization (Schulman et al., 2015) algorithm to satisfy constraints during training as well as after convergence. CPO is computationally expensive as it uses an approximation to the Fisher Information Matrix which requires many steps of conjugate gradient descent ($n_{cg}$ steps) followed by a backtracking line-search procedure ($n_{ls}$ steps) for each iteration, so it is more expensive by $\mathcal{O}(n_{cg} + n_{ls})$ per update. Furthermore, accurately estimating the curvature requires a large number of samples in each batch (Wu et al., 2017).

## 6 EXPERIMENTS

We empirically validate our approach on RL benchmarks to measure the performance of the agent with respect to the accumulated return and cost during training in the presence of neural-networks based function approximators. We compare our approach with the respective Unconstrained versions, and the Lyapunov-based approach (Chow et al., 2018; 2019) in each setting. Even though our formulation is based on the undiscounted case, we use discounting with $\gamma = 0.99$ for estimating the value functions in order to be consistent with the baselines. [7]

### 6.1 STOCHASTIC GRID WORLD

Motivated by the safety in navigation tasks, we first consider a stochastic 2D grid world (Leike et al., 2017; Chow et al., 2018). The agent (green cell in Fig. 1a) starts in the bottom-right corner, the safe region, and the objective is to move to the goal on the other side of the grid (blue cell). The agent can only move in the adjoining cells in the cardinal directions. It gets a reward of $+1000$ on reaching the goal, and a penalty of $-1$ at every timestep. Thus, the task is to reach the goal in the shortest amount of time. There are a number of pits in the terrain (red cells) that represent the safety constraint and the agent gets a cost of 10 on passing through any pit cell. Occasionally, with probability $p = 0.05$, a random action will be executed instead of the one selected by the agent. Thus, the task is to reach to the goal in the shortest amount of time, while passing through the red grids at most $d_0/10$ times. The size of the grid is $12 \times 12$ cells, and the pits are randomly generated for each grid with probability $\rho = 0.3$. The agent starts at $(12, 12)$ and the goal is selected uniformly on $(\alpha, 0)$, where $\alpha \sim U(0, 12)$. The threshold $d_0 = 20$ implies the agent can pass at most two pits. The maximum horizon is 200 steps, after which the episode terminates.

We use the action elimination procedure described in Sec 4.1 in combination with $n$-step SARSA (Rummery & Niranjan, 1994; Peng & Williams, 1994) using neural networks and multiple synchronous agents as in Mnih et al. (2016). We use $\epsilon$-greedy exploration. The results are shown in Fig. 1 (more experimental details can be found in Appendix H). We observe that the agent is able to respect the safety constraints more adequately than the Lyapunov-based method, albeit at the expense of some decrease in return, which is the expected trade-off for satisfying the constraints.

### 6.2 MUJOCO BENCHMARKS

Based on the safety experiments in Achiam et al. (2017); Chow et al. (2019), we design three simulated robot locomotion continuous control tasks using the MuJoCo simulator (Todorov et al., 2012) and OpenAI Gym (Brockman et al., 2016): (1) **Point-Gather**: A point-mass agent ($S \subseteq \mathbb{R}^9, A \subseteq \mathbb{R}^2$) is rewarded for collecting the green apples and constrained to avoid the red bombs; (2) **Safe-Cheetah**: A bi-pedal agent ($S \subseteq \mathbb{R}^{18}, A \subseteq \mathbb{R}^6$) is rewarded for running at high speed, but at the same time constrained by a speed limit; (3) **Point-Circle**: The point-mass agent ($S \subseteq \mathbb{R}^9, A \subseteq \mathbb{R}^2$) is rewarded for running along the circumference of a circle in counter-clockwise direction, but is constrained to stay within a safe region smaller than the radius of the circle.

We integrate our method on top of the A2C algorithms (Mnih et al., 2016) and PPO (Schulman et al., 2017), using the procedure described in Section 4.2. More details about the tasks and network architecture can be found in the Appendix I. Algorithmic details can be found in Appendix J. The results with A2C are shown in Fig. 2 and the results with PPO are shown in Fig. 3. We observe that

---

[7]In practice, the starting states in the episode are unlikely to be distributed according to the stationary distribution $\eta^{\pi}$. We still use the initial trajectories to update the estimates nonetheless in our experiments, but we use $n$-step updates.

our Safe method is able to respect the safety constraint throughout most of the learning, and with much greater degree of compliance than the Lyapunov-based method, especially when combined with A2C. The one case where the Safe method fails to respect the constraint is in Point-Circle with PPO (Fig. 3(c)). Upon further examination, we note that the training in this scenario has one of two outcomes: some runs end with the learner in an infeasible set of states from which it cannot recover; other runs end in a good policy that respects the constraint. We discuss solutions to overcome this in the final section.

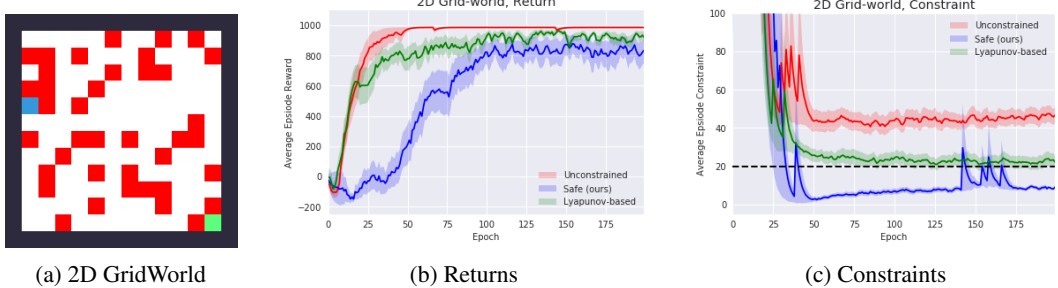

| (a) 2D GridWorld | (b) Returns | (c) Constraints |

Figure 1: (a) Example of a gridworld environment. (b,c) Performance over the training for Unconstrained (red), Lyapunov-based (green), and our method (blue) all trained with n-step SARSA on 2D GridWorld task over 20 random seeds. The x-axis is the number of episodes in thousands. The dotted black line in (c) denotes the constraint threshold, $d_0$. The bold line represents mean, and the shaded region denotes 80% confidence-intervals.

## 7 DISCUSSION

We present a method for solving constrained MDPs that respects trajectory-level constraints by converting them into state dependent value-based constraints, and show how the method can be used to handle safety limitations in both discrete and continuous spaces. The main advantage of our approach is that the optimization problem is more easily solved with value-based constraints, while providing similar guarantees and requiring less approximations. The empirical results presented show that our approach is able to solve the tasks with good performance while maintaining safety throughout training. It is important to note that there is a fundamental trade-off between exploration and safety. It is impossible to be 100% safe without some knowledge; in cases where that knowledge is not provided a priori, it must be acquired through exploration. We see this in some of our results (Gridworld, Point-Circle) where our safe policy goes above the constraint in the very early phases of training (all our experiments started from a random policy). We note that the other methods also suffer from this shortcoming. An open question is how to provide initial conditions or a priori knowledge, to avoid this burn-in phase. Another complementary strategy to explore is for cases where an agent is stuck in an unsafe or infeasible policy space, where a recovery method (trained by purely minimizing the constraints) could be useful to help the agent recover (Achiam et al., 2017; Chow et al., 2019).

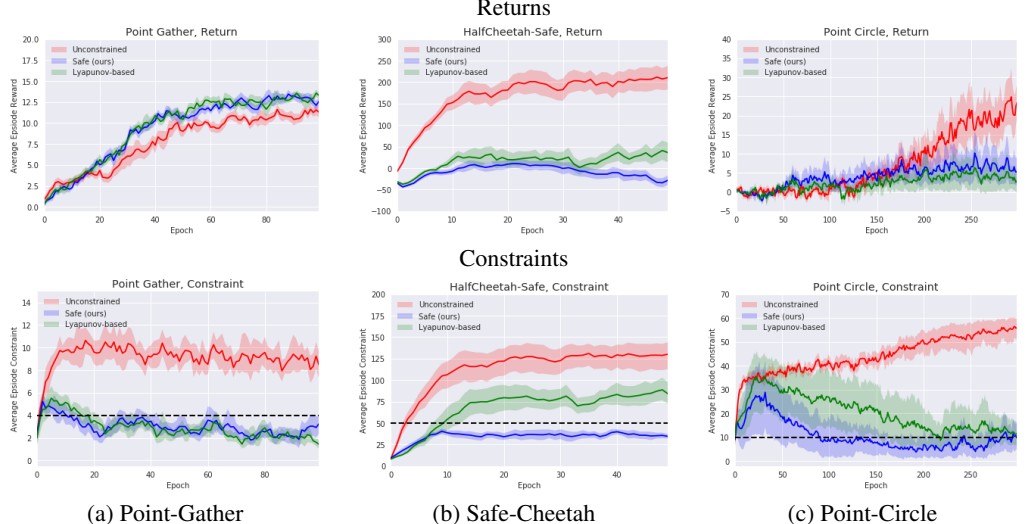

Figure 2: A2C Performance over the training for Unconstrained (red), Lyapunov-based (green), and our method (blue) all trained with A2C on MuJoCo tasks over 10 random seeds. The x-axis is the number of episodes in thousands. The dotted black line denotes $d_0$. The bold line represents the mean, and the shaded region denotes the 80% confidence-intervals.

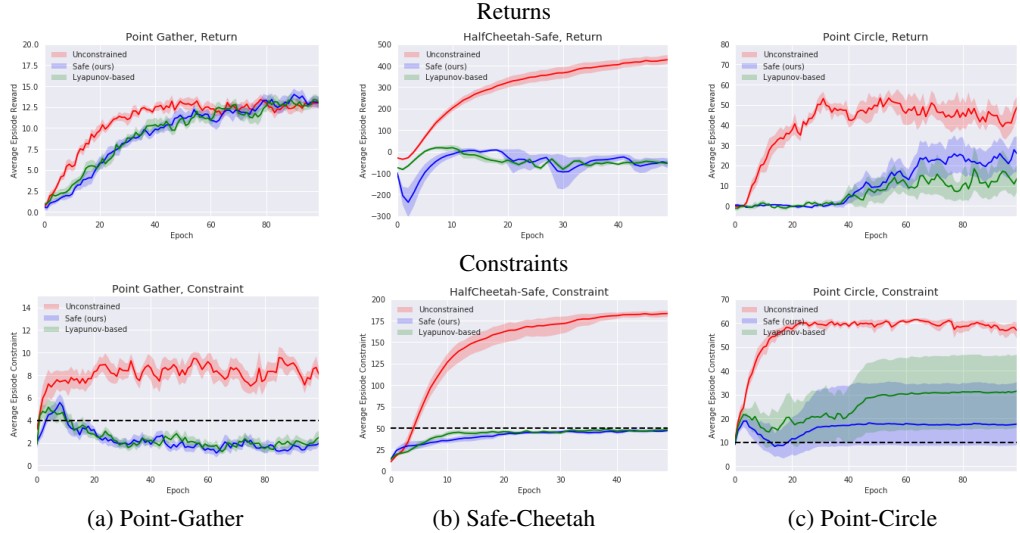

Figure 3: PPO Performance over the training for Unconstrained (red), Lyapunov-based (green), and our method (blue) all trained with PPO on MuJoCo tasks over 10 random seeds. The x-axis is the number of episodes in thousands, and y-axis denotes the undiscounted accumulated returns. The dotted black line denotes $d_0$. The bold line represents the mean, and the shaded region denotes the 80% confidence-intervals.

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

## A    REPRODUCIBILITY CHECKLIST

We follow the reproducibility checklist (Pineau, 2018) and point to relevant sections explaining them here. For all algorithms presented, check if you include:

- **A clear description of the algorithm.** The algorithms are explained in Sec. J. Any additional details for Discrete methods are provided in Sec. 4.1, and for continuous Sec. 4.2.

- **An analysis of the complexity (time, space, sample size) of the algorithm.** The analytical solution in Eq. (5) consists of a few vector arithmetic and relu operator and as such has the same complexity as the baselines. For the discrete case, with deterministic policies the solution again can be implemented as part of the computation graph, consisting of basic vector arithmetic operations, and has very little additional overhead. For discrete actions with stochastic policies, one needs to sovle the LP problem in (SPI). In that case the complexity is same as the baseline safe-methods (Lyapunov), and is higher than the unconstrained versions. In terms of computation time (for Deep-RL experiments) the newly proposed algorithms are almost identical to the baselines due to its parallelizable nature. We do not make any claims about the sample complexity.

- **A link to a downloadable source code, including all dependencies.** The code will be made available after the acceptance of the paper.

For any theoretical claim, check if you include:

- **A statement of the result.** See the main paper for all the claims we make. Additional details are provided in the Appendix.

- **A clear explanation of any assumptions.** See the main paper for all the assumptions.

- **A complete proof of the claim.** See the main paper. The cross references to the proofs in the Appendix have been included in the main paper.

For all figures and tables that present empirical results, check if you include:

- **A complete description of the data collection process, including sample size.** For the base agent we standard benchmarks provided in OpenAI Gym (Brockman et al., 2016), and rllab (Duan et al., 2016). We use the code from Achiam et al. (2017) for building the Point-Circle and Point-Gather environments.

- **A link to downloadable version of the dataset or simulation environment.** See: github.com/openai/gym for OpenAI Gym benchmarks, github.com/jachiam/cpo for rllab based Circle and Gather environments.

- **An explanation of how samples were allocated for training / validation / testing.** We do not use a split as we run multiple runs over random seeds to examine the optimization performance.

- **An explanation of any data that were excluded.** NA

- **The range of hyper-parameters considered, method to select the best hyper-parameter configuration, and specification of all hyper-parameters used to generate results.** The default hyper-parameters for the MuJoCo baselines are taken from Kostrikov (2018). The ranges and parameters for Grid experiments are described in Sec. H, and for MuJoCo are described in Sec. I.

- **The exact number of evaluation runs.** The number of evaluation runs is mentioned in the caption corresponding to each result.

- **A description of how experiments were run.** See Experiments Sec. 6 in the main paper and in the Appendix Sec. H and Sec. I.

- **A clear definition of the specific measure or statistics used to report results.** Undiscounted return and cost using the current policy over the horizon are plotted after every 1000 episodes are plotted. We use a linear-filter with 0.7 weight for smoothing. We use the smooting algorithm provided by TensorBoard (https://github.com/tensorflow/tensorboard).

- **Clearly defined error bars.** Standard error used in all cases.

- **A description of results with central tendency (e.g. mean) and variation (e.g. stddev).** The bold lines in the figure represent the mean, and the shaded region denotes the $80\%$ confidence interval.

- **A description of the computing infrastructure used.** We distribute all runs across 10 CPU nodes (Intel(R) Xeon(R) CPU E5-2650 v4) and 1 GPU (GP 100) per run for experiments.

# B    BACKWARD VALUE FUNCTIONS

We have the following result from Proposition 1 from Morimura et al. (2010). We give the proof too for the sake of completeness.

**Proposition B.1.** Let the forward Markov chain $\mathcal{M}(\pi)$ be irreducible and ergodic, i.e., has a stationary distribution. Then the associated backward Markov chain $\overleftarrow{\mathcal{B}}(\pi)$ is also ergodic and has the same unique stationary distribution as $\mathcal{M}(\pi)$:

$$\eta^\pi(x) = \overleftarrow{\eta}^\pi(x), \qquad (\forall x \in \mathcal{X})$$

where $\eta^\pi(x)$ and $\overleftarrow{\eta}^\pi(x)$ are the stationary distributions of $\mathcal{M}(\pi)$ and $\overleftarrow{\mathcal{B}}(\pi)$.

*Proof.* Multiply both sides of Eq. (1) by $\eta^\pi(x_t)$ and sum over all actions $a_{t-1} \in \mathcal{A}$ we obtain detailed balance like equations (with respect to time):

$$\overleftarrow{\mathcal{P}}^\pi(x_{t-1}|x_t)\eta^\pi(x_t) = \mathcal{P}^\pi(x_t, a_{t-1}|x_{t-1})\eta^\pi(x_{t-1}). \qquad (\forall x_{t-1} \in \mathcal{X}, x_t \in \mathcal{X})$$

Sum over all possible $x_t$ we have:

$$\sum_{x_t \in \mathcal{X}} \overleftarrow{\mathcal{P}}^\pi(x_{t-1}|x_t)\eta^\pi(x_t) = \eta^\pi(x_{t-1}).$$

The above equation indicates that $\overleftarrow{\mathcal{B}}(\pi)$ has same stationary distribution as $\mathcal{M}(\pi)$. In the matrix form the above equation can be written as $\eta\overleftarrow{P}^\pi = \eta$, that implies that $\eta$ is stationary distribution with $\overleftarrow{P}^\pi$ transition matrix. $\qquad \square$

## B.1    RELATION BETWEEN FORWARD AND BACKWARD MARKOV CHAINS AND BACKWARD VALUE FUNCTIONS

*Proof.* We use the technique of Proposition 2 of Morimura et al. (2010) to prove this. Using the Markov property and then substituting Eq. (1) for each term we have:

$$\overleftarrow{\mathcal{P}}^\pi(x_{t-1}, a_{t-1}, \ldots, x_{t-K}, a_{t-K}|x_t) = \overleftarrow{\mathcal{P}}^\pi(x_{t-1}, a_{t-1}|x_t)\ldots\overleftarrow{\mathcal{P}}^\pi(x_{t-K}, a_{t-K}|x_{t-K+1}),$$
$$= \frac{\mathcal{P}^\pi(x_t, a_{t-1}|x_{t-1})\ldots\mathcal{P}^\pi(x_{t-K+1}, a_{t-K}|x_{t-K})\eta^\pi(x_{t-K})}{\eta^\pi(x_t)},$$
$$\propto \mathcal{P}^\pi(x_t, a_{t-1}|x_{t-1})\ldots\mathcal{P}^\pi(x_{t-K+1}, a_{t-K}|x_{t-K})\eta^\pi(x_{t-K}).$$

This proves the proposition for finite $K$. Using the Prop. B.1, $K \to \infty$ case is proven too:

$$\lim_{K \to \infty} \mathbb{E}_{\overleftarrow{\mathcal{B}}(\pi)}\left[\sum_{k=0}^{K} d(x_{t-k})|x_t\right] = \lim_{K \to \infty} \mathbb{E}_{\mathcal{M}(\pi)}\left[\sum_{k=0}^{K} d(x_{t-k})|x_t, \eta^\pi(x_{t-K})\right],$$
$$= \sum_{x \in \mathcal{X}}\sum_{a \in \mathcal{A}} \pi(a|x)\eta^\pi(x)d(x).$$

$\qquad \square$

### B.2 TD FOR BVF

*Proof.* We use the same technique from Stochastic Shortest Path dynamic programming (Bertsekas et al., 1995, Vol 2, Proposition 1.1) to prove the above proposition. The general outline of the proof is given below, for more details we refer the reader to the textbook.

We have,

$$\overleftarrow{\mathcal{T}}^\pi \overleftarrow{V} = d + \overleftarrow{P}^\pi \overleftarrow{V}. \qquad \text{(Eq. (4) in matrix notation)}$$

Using induction argument, we have for all $\overleftarrow{V} \in \mathbb{R}^n$ and $k \geq 1$, we have:

$$\left(\overleftarrow{\mathcal{T}}^\pi\right)^k \overleftarrow{V} = \left(\overleftarrow{P}^\pi\right)^k \overleftarrow{V} + \sum_{m=0}^{k-1} \left(\overleftarrow{P}^\pi\right)^m d,$$

Taking the limit, and using the result, $\lim_{k\to\infty} \left(\overleftarrow{P}^\pi\right)^k \overleftarrow{V} = 0$, regarding proper policies from Bertsekas et al. (1995, Vol 2, Equation 1.2), we have:

$$\lim_{k\to\infty} \left(\overleftarrow{\mathcal{T}}^\pi\right)^k \overleftarrow{V} = \lim_{k\to\infty} \sum_{m=0}^{k-1} \left(\overleftarrow{P}^\pi\right)^m d = \overleftarrow{V}^\pi,$$

Also we have by definition:

$$\left(\overleftarrow{T}^\pi\right)^{k+1} \overleftarrow{V} = d + \overleftarrow{P}^\pi \left(\overleftarrow{T}^\pi\right)^k \overleftarrow{V},$$

and by taking the limit $k \to \infty$, we have:

$$\overleftarrow{V}^\pi = d + \overleftarrow{P}^\pi \overleftarrow{V}^\pi,$$

which is equivalent to,

$$\overleftarrow{V}^\pi = \overleftarrow{\mathcal{T}}^\pi \overleftarrow{V}^\pi.$$

To show uniqueness, note that if $\overleftarrow{V} = \overleftarrow{\mathcal{T}}^\pi \overleftarrow{V}$, then $\overleftarrow{V} = \left(\overleftarrow{\mathcal{T}}^\pi\right)^k \overleftarrow{V}$ for all $k$ and letting $k \to \infty$ we get $\overleftarrow{V} = \overleftarrow{V}^\pi$.

$\square$

## C  VALUE-BASED CONSTRAINT LEMMA

**Lemma C.1.** $\mathbb{E}\left[\sum_{k=t}^T d(x_k) \mid x_0, \pi\right] \leq \mathbb{E}_{x_t \sim \delta_{x_0}(P^\pi)^t}\left[V_{\mathcal{D}}^\pi(x_t)\right]$ and $\mathbb{E}\left[\sum_{k=0}^t d(x_k) \mid x_0, \pi\right] \leq \mathbb{E}_{x_k \sim \delta_{x_0}(P^\pi)^t}\left[\overleftarrow{V}_{\mathcal{D}}^\pi(x_k)\right]$

*Proof.* Follows since adding more steps to the trajectory (from $T - t$ steps to $T$) can only increase the expected total cost. $\mathbb{E}\left[\sum_{k=t}^T d(x_k) \mid x_0, \pi\right] = \delta_{x_0}(P^\pi)^t \left(\sum_{k=t}^T (P^\pi)^k\right) d \leq \delta_{x_0}(P^\pi)^t \left(\sum_{k=t}^{T+t} (P^\pi)^k\right) d = \mathbb{E}_{x_t \sim \delta_{x_0}(P^\pi)^t}\left[V_{\mathcal{D}}^\pi(x_t)\right]$. The backward case is analogous. $\square$

## D  PROPERTIES OF THE POLICY ITERATION (SPI)

**Theorem D.1.** Let $\sigma(x) := \text{TV}(\pi_{k+1}(\cdot|x), \pi_k(\cdot|x)) = (1/2) \sum_a |\pi_{k+1}(a|x) - \pi_k(a|x)|$ denote the total variation between policies $\pi_k(\cdot|x)$ and $\pi_{k+1}(\cdot|x)$. If the policies are updated sufficiently slowly and $\pi_k$ is feasible, then so is $\pi_{k+1}$. More specifically:

**(I)** If $\pi_k$ is feasible at $x_0$ and $\sigma(x) \leq \frac{d_0 - V_{\mathcal{D}}^{\pi_k}(x_0)}{2T^2 D_{\text{MAX}}}$ $\forall x$ then $\pi_{k+1}$ is feasible at $x_0$.

**(II)** If $\pi_k$ is feasible everywhere (i.e. $V_{\mathcal{D}}^{\pi_k}(x) \leq d_0$ $\forall x$) and $\sigma(x) \leq \frac{d_0 - V_{\mathcal{D}}^{\pi_k}(x)}{2T \max_{x'}\{d_0 - \overleftarrow{V}_{\mathcal{D}}^{\pi_k}(x') - d(x')\}}$ $\forall x$ then $\pi_{k+1}$ is feasible everywhere.

We note that the second case allows the policies to be updated in a larger neighborhood but requires $\pi_k$ to be feasible everywhere. By contrast the first item updates policies in a smaller neighbourhood but only requires feasibility at the starting state.

*Proof.* Similar to the analysis in Chow et al. (2018). We aim to show that $V_{\mathcal{D}}^{\pi_{k+1}}(x_0) \leq d_0$. For simplicity we consider $k = 0$, and by induction the other cases will follow. We write $P_0 = P^{\pi_0}, P_1 = P^{\pi_1}, \Delta(a|x) = \pi_1(a|x) - \pi_0(a|x)$, and $P_\Delta = \left[\sum_{a\in A} \Delta(a|x)P(x'|x,a)\right]_{\{x',x\}}$. Note that $(I - P_0) = (I - P_1 + P_\Delta)$, and therefore $(I - P_1 + P_\Delta)(I - P_0)^{-1} = I_{|\mathcal{X}|\times|\mathcal{X}|}$. Thus, we find

$$(I - P_0)^{-1} = (I - P_1)^{-1}(I_{|\mathcal{X}|\times|\mathcal{X}|} + P_\Delta(I - P_0)^{-1}).$$

Multiplying both sides by the cost vector $d$ one has

$$V_{\mathcal{D}}^{\pi_0}(x) = \mathbb{E}\left[\sum_{t=0}^{T} d(x_t) + \varepsilon(x_t) \mid \pi_1, x\right],$$

for each $x$, where $\varepsilon(x) = \sum_{a\in A} \Delta(a|x) \sum_{x'\in\mathcal{X}} P(x'|x,a)V_{\mathcal{D}}^{\pi_0}(x')$. Splitting the expectation, we have

$$V_{\mathcal{D}}^{\pi_1}(x) = V_{\mathcal{D}}^{\pi_0}(x) - \mathbb{E}\left[\sum_{t=0}^{T} \varepsilon(x_t) \mid \pi_1, x\right]$$

For case **(I)** we note that $V_{\mathcal{D}}^{\pi_0}(x') \leq D_{\text{MAX}}T$ and so $-2\sigma(x_t)D_{\text{MAX}}T \leq \varepsilon(x_t)$ $\forall x_t$. Using $\sigma(x_t) \leq (d_0 - V_{\mathcal{D}}^{\pi_k})/2D_{\text{MAX}}T^2$ gives $V_{\mathcal{D}}^{\pi_1}(x_0) \leq V_{\mathcal{D}}^{\pi_0}(x_0) - 2D_{\text{MAX}}T^2(d_0 - V_{\mathcal{D}}^{\pi_0}(x_0))/(2D_{\text{MAX}}T^2) = d_0$, i.e. $\pi_0$ is feasible at $x_0$.

For case **(II)** we note that $V_{\mathcal{D}}^{\pi_0}(x) \leq \max_{x'}\{d_0 - \overleftarrow{V}_{\mathcal{D}}^{\pi_0}(x') - d(x')\} =: \Theta$ since $\pi_0$ is feasible at every $x$. As before, we have $-2\sigma(x_t)\Theta \leq \varepsilon(x_t)$ $\forall x_t$ and so $V_{\mathcal{D}}^{\pi_1}(x) \leq V_{\mathcal{D}}^{\pi_0}(x) - 2\Theta T(d_0 - V_{\mathcal{D}}^{\pi_0}(x))/(2\Theta T) = d_0$ $\forall x$, i.e. $\pi_1$ is feasible everywhere. $\square$

**Theorem D.2.** Let $\pi_n$ and $\pi_{n+1}$ be successive policies generated be the policy iteration algorithm of (SPI). Then $V^{\pi_{n+1}} \geq V^{\pi_n}$.

*Proof.* Note that $\pi_{n+1}$ and $\pi_n$ are both feasible solutions of the LP (SPI). Since $\pi_{n+1}$ maximizes $V^\pi$ over all feasible solutions, the result follows. $\square$

# E  ANALYTICAL SOLUTION OF THE UPDATE - DISCRETE CASE

We follow the same procedure as (Chow et al., 2018, Section E.1) to convert the problem to its Shannon entropy regularized version:

$$\max_{\pi\in\Delta} \quad \pi(.|x)^T(Q(x,.) + \tau \log \pi(.|x)),$$
$$\text{s.t.} \quad \pi(.|x)^T Q_{\mathcal{D}}(x,.) + \overleftarrow{V}_{\mathcal{D}}^\pi(x) - d(x) \leq d_0, \tag{6}$$

where $\tau > 0$ is a regularization constant. Consider the Lagrangian problem for optimization:

$$\max_{\lambda\geq 0} \max_{\pi\in\Delta} \Gamma_x(\pi,\lambda) = \pi(.|x)^T(Q(x,.) + \lambda Q_{\mathcal{D}}(x,.) + \tau \log \pi(.|x)) + \lambda(d_0 + d(x) - \overleftarrow{V}(x))$$

From entropy-regularized literature (Neu et al., 2017), the inner $\lambda$-solution policy has the form:

$$\pi_{\Gamma,\lambda}^*(.|x) \propto \exp\left(-\frac{Q(x,.) + \lambda Q_{\mathcal{D}}(x,.)}{\tau}\right)$$

We now need to solve for the optimal lagrange multiplier $\lambda^*$ at $x$.

$$\max_{\lambda \geq 0} -\tau \log\text{-sum-exp}\left(-\frac{Q(x,.) + \lambda Q_{\mathcal{D}}(x,.)}{\tau}\right) + \lambda(d_0 + d(x) - \overleftarrow{V}_{\mathcal{D}}(x)),$$

where $\log\text{-sum-exp}(y) = \log \sum_a exp(y_a)$ is a convex function in $y$, and objective is a concave function of $\lambda$. Using KKT conditions, the $\nabla_\lambda$ gives the solution:

$$(d_0 + d(x) - \overleftarrow{V}_{\mathcal{D}}(x)) - \frac{\sum_a Q_{\mathcal{D}}(x,a) \exp\left(\left(-\frac{Q(x,a) + \lambda Q_{\mathcal{D}}(x,a)}{\tau}\right)\right)}{\sum_a \exp\left(\left(-\frac{Q(x,a) + \lambda Q_{\mathcal{D}}(x,a)}{\tau}\right)\right)} = 0$$

Using parameterization of $z = \exp(-\lambda)$, the above condition can be written as polynomial equation in $z$:

$$\sum_a \left(d_0 + d(x) - \overleftarrow{V}_{\mathcal{D}}(x) - Q_{\mathcal{D}}(x,a)\right) \cdot \left(\exp(-\frac{Q(x,a)}{\tau})\right) z^{\frac{Q_{\mathcal{D}}(x,a)}{\tau}} = 0$$

The roots to this polynomial will give $0 \leq z^*(x) \leq 1$, using which one can find $\lambda^*(x) = -\log(z^*(x))$. The roots can be found using the Newton's method. The final optimal policy of the entropy-regularized process is then:

$$\pi_\Gamma^* \propto \exp\left(-\frac{Q(x,\cdot) + \lambda^* Q_{\mathcal{D}}(x,\cdot)}{\tau}\right)$$

## F    EXTENSION OF SAFETY LAYER TO STOCHASTIC POLICIES WITH GAUSSIAN PARAMTERIZATION

Consider stochastic gaussian policies parameterized by mean $\mu(x;\theta)$ and standard-deviation $\sigma(x;\phi)$, and the actions sampled have the form $\mu(x;\theta) + \sigma(x;\phi)\epsilon$, where $\epsilon \sim \mathcal{N}(0,I)$ is the noise. Here, $< \mu(x;\theta), \sigma(x;\phi) >$ are both deterministic w.r.t. the parameters $\theta, \phi$ and $x$, and as such both of them together can be treated in the same way as deterministic policy ($\pi(x) = < \mu(x), \sigma(x) >$). The actual action sampled and executed in the environment is still stochastic, but we have moved the stochasticity fron the policy to the environment. This allows us to define and work with action-value functions $Q_{\mathcal{D}}(x, \mu_\pi(x), \sigma_\pi(x))$. In this case, the corresponding projected actions have the form $\mu' + \sigma'\epsilon$. The main objective of the safety layer (without the constraints) can be further simplified as:

$$\arg\min_{\mu',\sigma'} \mathbb{E}_{\epsilon \sim \mathcal{N}(0,I)} \left[\frac{1}{2} \|(\mu' + \sigma'\epsilon) - (\mu_\pi(x) + \sigma_\pi(x)\epsilon)\|^2\right]$$

$$\arg\min_{\mu',\sigma'} \mathbb{E}_{\epsilon \sim \mathcal{N}(0,I)} \left[\frac{1}{2} \|(\mu' - \mu_\pi(x)) + ((\sigma' - \sigma_\pi(x))\epsilon)\|^2\right]$$

$$\arg\min_{\mu',\sigma'} \frac{1}{2}\mathbb{E}_{\epsilon \sim \mathcal{N}(0,I)} \left[\|\mu' - \mu_\pi(x)\|^2 + \|(\sigma' - \sigma_\pi(x))\epsilon\|^2 + \underbrace{2 < \mu' - \mu_\pi(x), (\sigma' - \sigma_\pi(x))\epsilon >}_{=0,\text{due to linearity of expectation},\epsilon \sim \mathcal{N}(0,I)}\right]$$

$$\arg\min_{\mu',\sigma'} \frac{1}{2} \left(\|\mu' - \mu_\pi(x)\|^2 + \mathbb{E}_{\epsilon \sim \mathcal{N}(0,I)} \left[\|(\sigma' - \sigma_\pi(x))\epsilon\|^2\right]\right)$$

$$\arg\min_{\mu',\sigma'} \frac{1}{2} \left(\|\mu' - \mu_\pi(x)\|^2 + \|(\sigma' - \sigma_\pi(x))\|^2 \underbrace{\mathbb{E}_{\epsilon \sim \mathcal{N}(0,I)} \left[\|\epsilon\|^2\right]}_{=1,\text{second moment of } \epsilon}\right)$$

$$\arg\min_{\mu',\sigma'} \frac{1}{2} \left(\|\mu' - \mu_\pi(x)\|^2 + \|(\sigma' - \sigma_\pi(x))\|^2\right)$$

As both $\mu_\pi(.;\theta)$ and $\sigma_\pi(.;\phi)$ are modelled by independent set of parameters (different neural networks, usually) we can solve each of the safety layer problem independently, w.r.t. only those parameters.

# G ANALYTICAL SOLUTION IN SAFETY LAYER

The proof is similar to the proof of the Proposition 1 of Dalal et al. (2018). We have the following optimization problem:

$$\arg\min_{\mu}\left[\frac{1}{2}\left\|(\mu - \mu_\pi(x))\right\|^2\right],$$
$$\texttt{s.t.}\quad \overleftarrow{V}_{\mathcal{D}}^{\pi}(x) - d(x) + Q_{\mathcal{D}}^{\pi}(x, \mu_\pi(x)) + (\mu - \mu_\pi(x))^T(\nabla Q_{\mathcal{D}}^{\pi}(x, \mu)|_{\mu=\mu_\pi(x)}) \le d_0$$

As the objective function and constraints are convex, and the feasible solution, $\mu^*, \lambda^*$, should satisfy the KKT conditions. We define $\epsilon(x) = (d_0 + d(x) - \overleftarrow{V}_{\mathcal{D}}^{\pi}(x) - Q_{\mathcal{D}}^{\pi}(x, \mu_\pi(x)))$, and $g_{\mu,\mathcal{D}}(x) = \nabla Q_{\mathcal{D}}^{\pi}(x, u)|_{u=\mu_\pi(x)}$. Thus, we can write the Lagrangian as:

$$L(\mu, \lambda) = \frac{1}{2}\left\|(\mu - \mu_\pi(x))\right\|^2 + \lambda((\mu - \mu_\pi(x))^T g_{\mu,\mathcal{D}}(x) - \epsilon(x))$$

From the KKT conditions, we get:

$$\nabla_\mu L = \mu - \mu_\pi(x) + \lambda g_{\mu,\mathcal{D}}(x) = 0 \tag{7}$$
$$(\mu - \mu_\pi(x))^T g_{\mu,\mathcal{D}}(x) - \epsilon(x) = 0 \tag{8}$$

From Eq. (7), we have:

$$\mu^* = \mu_\pi(x) - \lambda^*(x) \cdot g_{\mu,D}(x) \tag{9}$$

Substituting Eq. (9) in Eq. (8), we get:

$$-\lambda^*(x) \cdot g_{\mu,D}(x)^T g_{\mu,D}(x) - \epsilon(x) = 0$$
$$\lambda^* = \frac{-\epsilon(x)}{g_{\mu,D}(x)^T g_{\mu,D}(x)}$$

When the constraints are satisfied ($\epsilon(x) > 0$), the $\lambda$ should be inactive, and hence we have $()^+$ operator, that is 0 for negative values.

# H DETAILS OF GRID-WORLD EXPERIMENTS

## H.1 ARCHITECTURE AND TRAINING DETAILS

We use one-hot encoding of the agent's location in the grid as the observation, i.e. $x$ is a binary vector of dimension $\mathbb{R}^{12\times12}$. The agent is trained for 200k episodes, and the current policy's performance is evaluated after every 1k episodes.

The same three layer neural network with the architecture is used for state encoding for all the different the estimators. The feed-forward neural network has hidden layers of size 64, 64, 64, and relu activations. For the state-action value based estimators, the last layer is a linear layer with 4 outputs, for each action. For value function based estimators the last layer is linear layer with a single output.

We use Adam Optimizer for training all the estimators. A learning rate of 1e-3 was selected for all the reward based estimators and a learning rate of 5e-4 was selected for all the cost based estimators. The same range of learning rate parameters for considered for all estimators i.e. {1e-5, 5e-5, 1e-4, 5e-4, 1e-3, 5e-3, 1e-2, 5e-2, 1e-1}.

We use n-step trajectory length in A2C with $n = 4$, i.e., trajectories of length $n$ were collected and the estimators were updated to used via the td-errors based on that. We use the number of parallel agents 20 in all the experiments. The range of parameters considered was $n \in \{1, 4, 20\}$. The same value of $n$ was used for all the baselines.

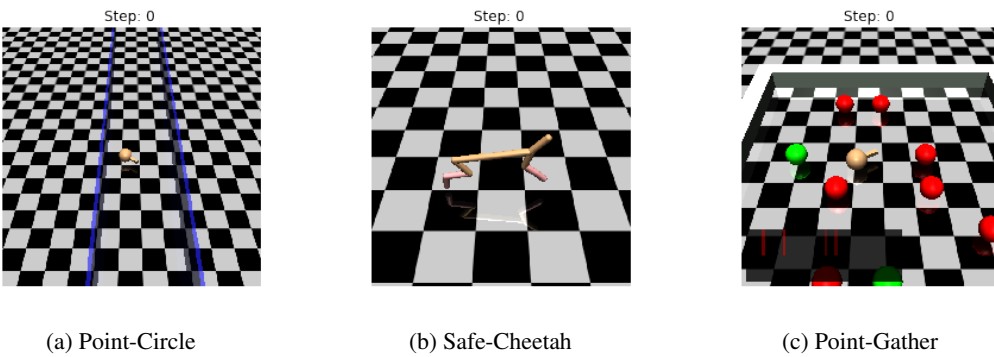

| (a) Point-Circle | (b) Safe-Cheetah | (c) Point-Gather |

Figure 4: MuJoCo Safety Environments

# I    DETAILS OF THE MUJOCO EXPERIMENTS

## I.1    ENVIRONMENTS DESCRIPTION

- **Point-Gather**: The environment (Fig.4c) is taken from Achiam et al. (2017), where the point mass agent gets a reward of $+10.0$ for collecting a green apple, and a cost of 1 for collecting a red bomb. Two apples and eight bombs are spawned randomly at the start of each episode. The constraints are defined over the nmber of bombs collected over the episode. Episode horizon is 15 and threshold $d_0 = 4$.

- **Safe-Cheetah**: This environment (Fig.4b) is taken from Chow et al. (2019). A bi-pedal agent (HalfCheetah-v0) is augmented with speed safety constraints. The agent gets the reward based on the speed with which it runs, and the constrain is define on the speed to be less than 1, i.e., it gets a constraint cost based on $\mathbb{1}[|v| > 1]$, where $v$ is the velocity at the state. The maximum length of the episode is 200 and the constraint threshold is $d0 = 50$.

- **Point-Circle**: This environment (Fig.4a) is taken from Achiam et al. (2017). The point-mass agent is rewarded for running along the circumference of a circle of radius 15 in counter-clockwise direction, with the reward and cost function:

$$R(s) = \frac{v^T[-y, x]}{1 + |\, \|[x, y]\|_2 - 15|},$$
$$C(s) = \mathbb{1}[|x| > 2.5],$$

where $x, y$ are coordinates in the plane and $v$ is the velocity. The length of the episode is 65 and the constraint threshold $d_0 = 10.0$.

## I.2    NETWORK ARCHITECTURE AND TRAINING DETAILS

The architecture and the training procedure is based on the open-source implementations (Kostrikov, 2018). All the value based estimators use a network architecture of 2 hidden layers of size 200, 50 hidden units with tanh non-linearity, followed by a linear layer with single output. For the actor, we model mean using a network architecture of 2 hidden layers of size 100, 50 hidden units with tanh non-linearity, followed by a linear layer with dimensions of the action-space and tanh non-linearity. For the $Q(x, \mu)$ we also a 2 layer neural network with 200, (50 + action-dimension) hidden units and tanh non-linearity. We concatenate the mean in the second layer, and add a linear layer with single output in the end.

Entropy regularization with $\beta = 0.001$ was used for all the experiments and the baselines. The trajectory length for different environments. For PPO GAE with $\lambda = 0.95$ was used for every algorithm. 20 parallel actors were used for every algorithm for each experiment. We searched the trajectory length hyper-parameter in the range 5,20,100 for every environment. We used the trajectory length of 1000 over which the samples are collected for PPO, for all environments. For the A2C experiments, for SafeCheetah trajectory length of 5 is used and for the rest 20 is used.

We use Adam Optimizer for training all the estimators. The learning rate of the critic is always 0.5 the learning rate of the actor. For the cost estimators, the same learning rate was used for forward and backward estimators. The same range of learning rate parameters for considered for all estimators i.e. {1e-5, 5e-5, 1e-4, 5e-4, 1e-3, 5e-3, 1e-2, 5e-2, 1e-1}.

### I.3 OTHER DETAILS

As we mentioned in Sec. 7, due to exploration the agent can potentially end up being in an infeasible policy space. To prevent that from happening a recovery policy (or safe-guard policy) (Achiam et al., 2017; Chow et al., 2019) is used to recover back to the feasible policy space. We run the experiments with and without the use of recovery policies (in the same procedure as the baselines), and chose the run that performs the best. We noticed that, empirically, for our approach recovery policies are only required for Point-Circle environments, as the agent has much more probability of being stuck in the constraint space.

In order to take error due to function approximation into account, Achiam et al. (2017) use cost-shaping to smooth out the sparse constraint, and Chow et al. (2019) use a relaxed threshold, i.e. $d_0 \cdot (1 - \delta)$, instead of $d_0$, where $\delta \in (0, 1)$. We run experiments with $\delta = \{0.0, 0.2\}$ for each algorithms, and use the best among them. We found that empirically, only for Safe-Cheetah $\delta = 0.2$ works better compared to $\delta = 0.0$.

## J ALGORITHM DETAILS

### J.1 n-STEP SYNCHRONOUS SARSA

The algorithm for n-step Synchronous SARSA is similar to the n-step Asynchronous Q-learning of Mnih et al. (2016), except that it uses SARSA instead of Q-learning, is synchronous, and instead of greedy maximization step of $\epsilon$-greedy we use (SPI). When working with discrete actions and deterministic policies, this can be solved as part of the computation-graph itself. The algorithm is presented in Alg. 1.

### J.2 A2C

In Actor Critic (Konda & Tsitsiklis, 2000) algorithms, the parameterized policy (actor) is denoted by $\pi(a|x; \theta)$, and is updated to minimizing the following loss:

$$L(\theta) = \mathbb{E}[-\log \pi(a_t|x_t; \theta)(r_t + \gamma V^\pi(x_{t+1} - V_{x_t}))]$$

The algorithm for A2C with Safety Layer given by Eq. (5) is similar to the Synchronous version of Actor-Critic (Mnih et al., 2016), except that it has estimates for the costs and safety-layer. Note that due to the projection property of the safety layer, it is possible to sample directly from the projected mean. Also, as the projection is a result of vector products and max, it is differentiable and and computed in-graph (via relu). The algorithm is presented in Alg. 2.

### J.3 PPO

The PPO algorithm build on top of the Actor-Critic algorithm and is very similar to Algorithm 2. The main difference is how the PPO loss for the actor is defined as:

$$L^{CLIP}(\theta) = \mathbb{E}[\min(\rho_t(\theta)A_t, clip(\rho_t(\theta), 1 - \epsilon, 1 + \epsilon)A_t)],$$

where the likelihood ration is $\rho_t(\theta) = \frac{\pi_\theta(a_t|x_t)}{\pi_{\theta_{old}}(a_t|x_t)}$, with $\pi_{old}$ being the policy parameters before the update, $\epsilon < 1$ is a hyper-parameters that controls the clipping and $A_t$ is the generalized advantage estimator:

$$A_t^{GAE(\lambda,\gamma)} = \sum_{k=0}^{T-1} (\lambda\gamma)^k \delta_{t+k}^{V^\pi},$$

---

**Algorithm 1** Synchronous n-step SARSA

---

**Input:** $\theta$ parameters for $Q(x, .; \theta)$, $\theta_{\mathcal{D}}$ parameters for $Q_{\mathcal{D}}(x, .; \theta_{\mathcal{D}})$, $\phi_{\mathcal{D}}$ parameters for $\overleftarrow{V}_{\mathcal{D}}(x; \phi_{\mathcal{D}})$; $\pi_0$ initial feasible policy.

**for** episode $e \in 1, ..., M$ **do**

    Add the initial state to the trajectory buffer $\tau \leftarrow \{x_0\}$

    $t \leftarrow 1$

    **while** $t < T$ **do**:

        $t_{start} \leftarrow t$

        **while** $t < t + n$ or $t == T$ **do**

            Select $a_t$ using (SPI), execute $a_t$, observe $x_{t+1}$ and reward $r_t$ and cost $d_t$.

            Add experiences to a buffer, i.e., $\tau \leftarrow (a_t, r_t, d_t, x_{t+1})$.

            $t \leftarrow t + 1$

        **end while**

        Calculate the next action for $x_{t+1}$ using the current policy estimates, $a_{t+1}$

        Bootstrap the targets:

$$R \leftarrow \begin{cases} 0 & \text{if } t == T \\ Q(x_{t+1}, a_{t+1}; \theta) & \text{otherwise} \end{cases}$$

$$R_{\mathcal{D}} \leftarrow \begin{cases} 0 & \text{if } t == T \\ Q_{\mathcal{D}}(x_{t+1}, a_{t+1}; \theta_{\mathcal{D}}) & \text{otherwise} \end{cases}$$

$$\overleftarrow{R}_{\mathcal{D}} \leftarrow \begin{cases} 0 & \text{if } t == 0 \\ \overleftarrow{V}(x_{t_{start}-1}; \phi_{\mathcal{D}}) & \text{otherwise} \end{cases}$$

                                            $\triangleright$ Calculate the targets for the transitions in buffer

    **for** $i \in \{t - 1, \ldots, t_{start}\}$ **do**

        $R \leftarrow r_i + \gamma R$

        $R_{\mathcal{D}} \leftarrow d_i + \gamma R_{\mathcal{D}}$

        Accumulate the gradients wrt $\theta, \theta_{\mathcal{D}}$:

$$d\theta \leftarrow d\theta + \frac{\partial(R - Q(x_i, a_i; \theta))^2}{\partial \theta}$$

$$d\theta_{\mathcal{D}} \leftarrow d\theta_{\mathcal{D}} + \frac{\partial(R_{\mathcal{D}} - Q_{\mathcal{D}}(x_i, a_i; \theta_{\mathcal{D}}))^2}{\partial \theta_{\mathcal{D}}}$$

    **end for**

    **for** $i \in \{t_{start}, \ldots, t\}$ **do**

        $\overleftarrow{R}_{\mathcal{D}} \leftarrow d_i + \gamma \overleftarrow{R}_{\mathcal{D}}$

        Accumulate the gradients wrt $\phi_{\mathcal{D}}$:

$$d\phi_{\mathcal{D}} \leftarrow d\phi_{\mathcal{D}} + \frac{\partial(\overleftarrow{R}_{\mathcal{D}} - \overleftarrow{V}_{\mathcal{D}}(x_i; \phi_{\mathcal{D}}))^2}{\partial \phi_{\mathcal{D}}}$$

    **end for**

        Do synchronous batch update with the accumulated gradients to update $\theta, \theta_{\mathcal{D}}, \phi_{\mathcal{D}}$ using $d\theta, d\theta_{\mathcal{D}}, d\phi_{\mathcal{D}}$.

    **end while**

    Empty the trajectory buffer, $\tau$

**end for**

---

---

**Algorithm 2** Synchronous A2C with Safety Layer

---

**Input:** $\theta$ parameters for $\pi(x; \theta)$, $\phi$ the parameters for $V(x; \phi)$, $\theta_{\mathcal{D}}$ parameters for $Q_{\mathcal{D}}(x, \mu; \theta_{\mathcal{D}})$, $\phi_{\mathcal{D}}$ parameters for $\overleftarrow{V}_{\mathcal{D}}(x; \phi_{\mathcal{D}})$;
**for** episode $e \in 1, ..., M$ **do**
    Add the initial state to the trajectory buffer $\tau \leftarrow \{x_0\}$
    $t \leftarrow 1$
    **while** $t < T$ **do**:
        $t_{start} \leftarrow t$
        **while** $t < t + n$ or $t == T$ **do**
            Select $a_t$ using sampling from the projected mean $\mu_t$ via the safety layer Eq.(5), execute $a_t$, observe $x_{t+1}$ and reward $r_t$ and cost $d_t$.
            Add experiences to a buffer, i.e., $\tau \leftarrow (a_t, \mu_t, r_t, d_t, x_{t+1})$.
            $t \leftarrow t + 1$
        **end while**
        Calculate the next mean for $x_{t+1}$ using the current policy estimates, $\mu_{t+1}$
        Bootstrap the targets:

$$R \leftarrow \begin{cases} 0 & \text{if } t == T \\ V(x_{t+1}, a_{t+1}; \phi) & \text{otherwise} \end{cases}$$

$$R_{\mathcal{D}} \leftarrow \begin{cases} 0 & \text{if } t == T \\ Q_{\mathcal{D}}(x_{t+1}, \mu_{t+1}; \theta_{\mathcal{D}}) & \text{otherwise} \end{cases}$$

$$\overleftarrow{R}_{\mathcal{D}} \leftarrow \begin{cases} 0 & \text{if } t == 0 \\ \overleftarrow{V}(x_{t_{start}-1}; \phi_{\mathcal{D}}) & \text{otherwise} \end{cases}$$

                                            ▷ Calculate the targets for the transitions in buffer
        **for** $i \in \{t - 1, \ldots, t_{start}\}$ **do**
            $R \leftarrow r_i + \gamma R$
            $R_{\mathcal{D}} \leftarrow d_i + \gamma R_{\mathcal{D}}$
            Accumulate the gradients w.r.t. $\theta, \phi, \theta_{\mathcal{D}}$:

$$d\theta \leftarrow d\theta + \nabla_\theta \log \pi(a_i \mid x_i; \theta)(R - V(x_i; \phi))$$

$$d\phi \leftarrow d\phi + \frac{\partial(R - V(x_i \phi))^2}{\partial \phi}$$

$$d\theta_{\mathcal{D}} \leftarrow d\theta_{\mathcal{D}} + \frac{\partial(R_{\mathcal{D}} - Q_{\mathcal{D}}(x_i, \mu_i; \theta_{\mathcal{D}}))^2}{\partial \theta_{\mathcal{D}}}$$

        **end for**
        **for** $i \in \{t_{start}, \ldots, t\}$ **do**
            $\overleftarrow{R}_{\mathcal{D}} \leftarrow d_i + \gamma \overleftarrow{R}_{\mathcal{D}}$
            Accumulate the gradients wrt $\phi_{\mathcal{D}}$:

$$d\phi_{\mathcal{D}} \leftarrow d\phi_{\mathcal{D}} + \frac{\partial(\overleftarrow{R}_{\mathcal{D}} - \overleftarrow{V}_{\mathcal{D}}(x_i; \phi_{\mathcal{D}}))^2}{\partial \phi_{\mathcal{D}}}$$

        **end for**
        Do synchronous batch update with the accumulated gradients to update $\theta, \phi, \theta_{\mathcal{D}}, \phi_{\mathcal{D}}$ using $d\theta, d\phi, d\theta_{\mathcal{D}}, d\phi_{\mathcal{D}}$.
    **end while**
    Empty the trajectory buffer, $\tau$
**end for**

---

where $T$ is the maxmimum number of timestamps in an episode trajectory, and $\delta_j$ denotes the TD error at $j$. The value function is updated using the $\gamma\lambda$-returns from the GAE:

$$L(\phi) = \mathbb{E}[(V^\pi(x; \phi) - (V^\pi(x; \phi_{old}) + A_t))^2].$$

Similar to the the forward value estimates the backward value estimates are defined in the similar sense. One way to think of it is to assume the trajectories are reversed and we are doing the regular GAE estimation for the value functions.

The GAE updates for the regular value function can be seen in the $\lambda$-operator form as:

$$\mathcal{T}_\lambda^\pi v^\pi = (I - \gamma\lambda P^\pi)^{-1}(r^\pi + \gamma P^\pi v^\pi - v^\pi) + v^\pi.$$

In similar spirit it can be shown that the $\lambda$-operator for SARSA has the form:

$$\mathcal{T}_\lambda^\pi q^\pi = (I - \lambda\gamma P^\pi)^{-1}(\mathcal{T}^\pi q^\pi - q^\pi) + q^\pi,$$

where $(\mathcal{T}^\pi q^\pi - q^\pi)$ denotes the TD error. Thus, the GAE estimates can be applied for the Q-functions in the similar form, i.e.

$$B_t^{GAE(\lambda,\gamma)} = \sum_{k=0}^{T-1} (\lambda\gamma)^k \delta_{t+k}^{Q_\mathcal{D}^\pi},$$

$$L(\theta_\mathcal{D}) = \mathbb{E}[(Q_\mathcal{D}^\pi(x, a; \theta_\mathcal{D}) - (Q_\mathcal{D}^{\theta_{\mathcal{D}}}(x, a; \theta_{\mathcal{D}_{old}}) + B_t))^2].$$

