# OpenReview forum: "Constrained Markov Decision Processes via Backward Value Functions"
_ICLR.cc/2020/Conference — Reject_

### Official Review · AnonReviewer1 · 2019-10-14
**Official Blind Review #1**

**Rating:** 3

**Review:**

The paper approaches the CMDP problem, in which one wishes to learn a max return policy subject to trajectory-based constraints.  The paper proposes a technique based on the introduced concept of "backward value functions".  These functions satisfy a sort of Bellman equation.  The paper proposes a safe policy improvement step based on these value functions, with theoretical guarantees on the safety of the resulting policy.  The method is evaluated on gridworlds and mujoco tasks, showing good performance.

The paper provides nice results some intriguing ideas, although after reading I am left with several questions as to the details of the method and its place relative to previous works.

-- How is the backwards value function learned exactly?  Since it relies on a backwards Bellman operator, it seems to necessitate on-policy samples.  Is the function learned in an off-policy fashion somehow?
-- The problem solved by SPI (constraint for each state) seems much more conservative than the constraint at the bottom of page 4 (single constraint based on average over all states). In it current form, the algorithm appears to be overly conservative, to the point that it may converge to a policy very far from optimal.
-- I am confused by the assumption on policies updating "sufficiently slowly".  Why is this needed?  Also, how does this relate to CPO (Achiam, et al.)?  It would appear that the methodology of CPO is more desirable, since it only requires the "sufficiently slow" assumption without the additional backward-value constraints.

**Experience Assessment:**

I have published one or two papers in this area.

**Review Assessment: Checking Correctness Of Derivations And Theory:**

I assessed the sensibility of the derivations and theory.

**Review Assessment: Checking Correctness Of Experiments:**

I assessed the sensibility of the experiments.

**Review Assessment: Thoroughness In Paper Reading:**

I read the paper at least twice and used my best judgement in assessing the paper.

---

> ### Author Response · Authors · 2019-11-08
> **Thanks for your comments**
>
> We thank the reviewer for providing the feedback.  We address the concerns raised by the reviewer here:
>
> 1) As we mention in the main text (p. 1 Abstract, p. 2 Sec. 1), the entire approach is on-policy. As such, the backward value functions are also calculated in the on-policy manner. We have Proposition 3.1 that justifies the use of samples from a forward Markov chain to estimate the backward value functions. In Proposition 3.2 we show the existence of fixed point of the backward value function for policy evaluation.
>
> 2) Yes, we remark that we are working on a more conservative (upper bound) of the original problem (p. 4, Sec 3.3). As we mentioned in the main text(p. 5, Sec 3.3), the theoretical analysis of the sub-optimality gap remains an interesting question left for future work.  However, the empirical results in Section 6 show that our approach works well in practice.
>
> 3)  The slow update assumption is required to satisfy the consistent feasibility property of the safe policy iteration procedure. This assumption can be integrated with the conservative policy improvement framework by moving it to the constraints, similar to CPO and TRPO, however, the resulting formulation will lead to an alternate form of the CPO. Solving the formulation with a monotonic policy improvement framework (Kakade et al, 2002), like CPO or TRPO, requires knowledge of the Fisher Information Matrix (FIM) to claim any guarantees. Moreover, for empirical purposes, as the FIM is not computationally tractable, an approximate estimate is used. Instead, our approach presents an alternate way to work with the analytical solutions of the state-dependent constraints.
>
>
> Essentially, the slow update assumption can be moved to the constraints, to get a formulation similar to the CPO, however, it comes with a trade-off of working with an approximate FIM (that is computationally expensive) instead of analytical solutions provided by the state-dependent constraints. However, this is not true the other way around.
>
> References:
>
> Sham Kakade and John Langford. Approximately optimal approximate reinforcement learning. In International Conference on Machine Learning, volume 2, 2002.

---

### Official Review · AnonReviewer3 · 2019-10-23
**Official Blind Review #3**

**Rating:** 8

**Review:**

This paper presents a new approach for solving Constrained MDPs. Because the cost constraint is cumulative, the best action depends on the cumulative cost so far. They address this issue by learning a backward value function of the estimated cumulative cost so far. Their theoretical results show that the same properties for forward value functions hold here for backwards ones. They are then able to use the forward and backward cost estimates to constraint the actions selection, by adding a safety layer to the algorithm. The results show that the method does a better job of meeting safety constraints than the Lyapunov based method.

The backward value function idea is a nice novel way of addressing the cumulative cost constraint problem.

The paper is clearly written and the results are nice.

The biggest issue with this paper is that too much material has been pushed to the appendix. I think at the least some of the details of the actual implementation with PPO or A2C should be moved into the main text.

For the empirical results, it would be great to see something about the computation or wall clock time required. Does this method run more quickly than the Lyapunov based method?

**Experience Assessment:**

I have published one or two papers in this area.

**Review Assessment: Checking Correctness Of Derivations And Theory:**

I assessed the sensibility of the derivations and theory.

**Review Assessment: Checking Correctness Of Experiments:**

I carefully checked the experiments.

**Review Assessment: Thoroughness In Paper Reading:**

I read the paper at least twice and used my best judgement in assessing the paper.

---

> ### Author Response · Authors · 2019-11-08
> **Thanks for your comments**
>
> We thank the reviewer for providing the feedback.  We address the concerns raised by the reviewer here:
>
> - Given the length flexibility afforded at ICLR, we will be glad to move details for the deep RL methods to the main text for the final version.
>
> - We mention this briefly in the reproducibility checklist (Sec. A, Appendix) that the computation time (for Deep-RL experiments) for the newly proposed algorithms are almost identical to the baselines due to its parallelizable nature. We can add this to the main text also.

---

### Official Review · AnonReviewer2 · 2019-10-24
**Official Blind Review #2**

**Rating:** 3

**Review:**

In this work, the authors studied solving the CMDP problem, in particular to model the constraint propagation using the idea of the backward value functions. Using the time-reversed probability property, they first established Bellman optimality conditions for the backward value function, which accounts for the tail expected return, conditioned on the final state.
Utilizing  the notion of backward value function, they further use that to model the constraint in the CMDPs, and proposed several safe policy iteration algorithms for both cases with discrete actions and with continuous actions. Experiments show that this method has better constraint guarantees than the state-of-the-art algorithms, such as the Lyapunov approach.

The idea of using backward value functions to model constraints in CMDP is interesting and so far I have not seen it in other places. The algorithms developed with this approach also appear to work reasonably well (especially in constraint guarantees) on benchmark domains such as gridworld and mujoco control tasks. The authors also provide several properties such as consistent feasibility and policy improvement, similar to the Lyapunov method,  and derive several versions of safe policy optimization algorithms. However, I found the intuition of using backward value function rather unclear. Unlike the Lyapunov function, which attempts to estimate a near-optimal "remaining constraint budget" w.r.t. CMDP constraint, what is the motivation behind using the backward value. Does the backward probability have any connections to occupation measures? Without similar motivations it is unclear how close the performance of the policy computed from this backward value approach with that of an optimal CMDP policy. Also unlike the Lyapunov approach in Chow'18, the consistent feasibility  property in this work appears to be more restricted as it is only limited to a slow policy update. Finally the experimental results look promising, but it would be great to also compare with other state-of-the-art results such as Lagrangian method, and conservative policy improvement (such as CPO).



**Experience Assessment:**

I have published in this field for several years.

**Review Assessment: Checking Correctness Of Derivations And Theory:**

I assessed the sensibility of the derivations and theory.

**Review Assessment: Checking Correctness Of Experiments:**

I assessed the sensibility of the experiments.

**Review Assessment: Thoroughness In Paper Reading:**

I read the paper at least twice and used my best judgement in assessing the paper.

---

> ### Author Response · Authors · 2019-11-08
> **Thanks for your comments**
>
> We thank the reviewer for providing the feedback.  We address the concerns raised by the reviewer here:
>
> 1) Intuition
>
> Indeed, the main intuition for using backward value functions is to use them for estimating the remaining constraint budget for a given policy. In the safe policy improvement (Eq. SPI), the quantity of the LHS of the constraint denotes the expected constraint accumulated over the trajectory (expected_cost_future + expected_cost_past). An alternate way to look at the constraint in Eq. SPI is to rearrange the terms, moving the terms from LHS to RHS, we get:
> $$ d_0 - (<\pi(.|x) Q^{\pi_k}_{D}(x, .)> + V_{D}^{\leftarrow \pi_k} - d(x)) >= 0 $$
> The above equation can also be seen as:
> $$ \text{Remaing_constraint_budget} >= 0 $$ .
> The policy improvement step is then performed which takes into account the remaining constraint budget. In this way, we use the backward value functions to provide an alternate way to estimate the remaining constraint budget.
>
> In relation to occupation measures, the backward value functions are related to occupation measures in a similar sense that the forward value functions are related to the occupancy measure.  The forward value functions can be written as future occupation measure times the expected reward, and in a similar spirit, the backward value function can be interpreted as past occupation measure times the expected reward. Intuitively, the combination of forward and backward value functions gives the occupation measure for the entire trajectory.
>
> We hope that this gives more insights into the intuition of using backward value functions. We will incorporate this into the paper.
>
>
> 2)  Consistent feasibility property
>
> At first glance, it might appear that the consistent feasibility of our work is more restricted than the Lyapunov approach (Chow et al. 2018), but there is a fundamental difference here. For the property in Chow et al. (2018) to be valid, it relies on the assumption between the closeness of the baseline policy and the optimal policy (Assumption 1, Chow et al. 2018), i.e., $D_{TV}(\pi^{*}(.|x)||\pi^{0}(.|x))$.  Whereas in our work, instead of the relation between starting baseline policy and the optimal policy, the slow policy update applies to the successive policies in the policy iteration procedure $D_{TV}(\pi^{k+1}(.|x)||\pi^{k}(.|x))$.
> Depending on the kind of problem, there are scenarios where this property is more desirable than the one in Chow et al. (2018), for instance, in the case when the initial baseline policy is random and we don’t have access to the optimal policy.
>
> 3) More experiments with Lagrangian and CPO
>
> We limited our comparison with the existing SOTA method based on the Lyapunov approach Chow et al. (2019). We have benchmarked our implementation of the Lyapunov baseline using the hyper-parameters provided by the authors. In the related work section, we mention the reasons for not comparing with Langrangian (no guarantees, difficult to optimize) and CPO methods (more computationally expensive).  We point the reviewer to the study related to the comparison between the Lyapunov approach and Langragian methods that can be found in Chow et al. (2019).    For completeness, we will aim to re-run these baselines for the final version.
>
>
> References:
>
> Chow, Yinlam, et al. "A lyapunov-based approach to safe reinforcement learning." Advances in Neural Information Processing Systems. 2018.
>
> Yinlam Chow, Ofir Nachum, Aleksandra Faust, Mohammad Ghavamzadeh, and Edgar Duenez Guzman. Lyapunov-based safe policy optimization for continuous control. arXiv preprint arXiv:1901.10031, 2019.

---

### Decision · Program_Chairs · 2019-12-19

**Decision:**

Reject

**Comment:**

The paper considers the setting of constrained MDPs and proposes using backward value functions to keep track of the constraints.

All reviewers agreed that the idea of backward value functions is interesting, but there were a few technical concerns raised, and the reviewers remained unconvinced after the rebuttal. In particular, there were doubts whether the method actually makes sense for the considered problem (the backward VF averaging constraints over all trajectories, instead of only considering the current one), and a concern about insufficient baseline comparisons.

I recommend rejection at this time, but encourage the authors to take the feedback into account, make the paper more crisp, and resubmit to a future venue.